# Conservation and divergence of myelin proteome and oligodendrocyte transcriptome profiles between humans and mice

**Vasiliki-Ilya Gargareta[1†], Josefine Reuschenbach[1†], Sophie B Siems[1†], Ting Sun[1*†], Lars Piepkorn[2,3], Carolina Mangana[1], Erik Späte[1], Sandra Goebbels[1], Inge Huitinga[4,5], Wiebke Möbius[1,6], Klaus-Armin Nave[1], Olaf Jahn[2,3*], Hauke B Werner[1*]**

[1]Department of Neurogenetics, Max Planck Institute for Multidisciplinary Sciences, Göttingen, Germany; [2]Neuroproteomics Group, Department of Molecular Neurobiology, Max Planck Institute for Multidisciplinary Sciences, Göttingen, Germany; [3]Translational Neuroproteomics Group, Department of Psychiatry and Psychotherapy, University Medical Center Göttingen, Georg-August-University, Göttingen, Germany; [4]University of Amsterdam, Swammerdam Institute for Life Sciences, Brain Plasticity Group, Amsterdam, Netherlands; [5]Neuroimmunology Group, Netherlands Institute for Neuroscience, Amsterdam, Netherlands; [6]Electron Microscopy Unit, Max Planck Institute for Multidisciplinary Sciences, Göttingen, Germany

**\*For correspondence:**
tsun@mpinat.mpg.de (TS);
jahn@mpinat.mpg.de (OJ);
hauke@mpinat.mpg.de (HBW)

[†]These authors contributed equally to this work

**Abstract** Human myelin disorders are commonly studied in mouse models. Since both clades evolutionarily diverged approximately 85 million years ago, it is critical to know to what extent the myelin protein composition has remained similar. Here, we use quantitative proteomics to analyze myelin purified from human white matter and find that the relative abundance of the structural myelin proteins PLP, MBP, CNP, and SEPTIN8 correlates well with that in C57Bl/6N mice. Conversely, multiple other proteins were identified exclusively or predominantly in human or mouse myelin. This is exemplified by peripheral myelin protein 2 (PMP2), which was specific to human central nervous system myelin, while tetraspanin-2 (TSPAN2) and connexin-29 (CX29/GJC3) were confined to mouse myelin. Assessing published scRNA-seq-datasets, human and mouse oligodendrocytes display well-correlating transcriptome profiles but divergent expression of distinct genes, including *Pmp2*, *Tspan2,* and *Gjc3*. A searchable web interface is accessible via www.mpinat.mpg.de/myelin. Species-dependent diversity of oligodendroglial mRNA expression and myelin protein composition can be informative when translating from mouse models to humans.

## Editor's evaluation

In this impressive article, the authors study the similarities and differences between the molecules that comprise the insulation that surrounds human brain nerve fibers (myelin), providing essential insight into how to interpret studies of myelin, from the perspective of different species. In all, this article provides a new resource that will be of interest to the myelin community as well as investigators examining the contributions of oligodendrocytes to human neurodegenerative disease.

**eLife digest** Like the electrical wires in our homes, the processes of nerve cells – the axons, thin extensions that project from the cell bodies – need to be insulated to work effectively. This insulation takes the form of layers of a membrane called myelin, which is made of proteins and fats and produced by specialized cells called oligodendrocytes in the brain and the spinal cord. If this layer of insulation becomes damaged, the electrical impulses travelling along the nerves slow down, affecting the ability to walk, speak, see or think. This is the cause of several illnesses, including multiple sclerosis and a group of rare genetic diseases known as leukodystrophies.

A lot of the research into myelin, oligodendrocytes and the diseases caused by myelin damage uses mice as an experimental model for humans. Using mice for this type of research is appropriate because of the ethical and technical limitations of experiments on humans. This approach can be highly effective because mice and humans share a large proportion of their genes. However, there are many obvious physical differences between the two species, making it important to determine whether the results of experiments performed in mice are applicable to humans. To do this, it is necessary to understand how myelin differs between these two species at the molecular level.

Gargareta, Reuschenbach, Siems, Sun et al. approached this problem by studying the proteins found in myelin isolated from the brains of people who had passed away and donated their organs for scientific research. They used a technique called mass spectrometry, which identifies molecules based on their weight, to produce a list of proteins in human myelin that could then be compared to existing data from mouse myelin. This analysis showed that myelin is very similar in both species, but some proteins only appear in humans or in mice. Gargareta, Reuschenbach, Siems, Sun et al. then compared which genes are turned on in the oligodendrocytes making the myelin. The results of this comparison reflected most of the differences and similarities seen in the myelin proteins.

Despite the similarities identified by Gargareta, Reuschenbach, Siems, Sun et al., it became evident that there are unexpected differences between the myelin of humans and mice that will need to be considered when applying results from mice research to humans. To enable this endeavor, Gargareta, Reuschenbach, Siems, Sun et al. have created a searchable web interface of the proteins in myelin and the genes expressed in oligodendrocytes in the two species.

## Introduction

Oligodendrocytes support axons in the central nervous system (CNS) of vertebrates both metabolically and by providing myelin sheaths, which enable rapid, saltatory impulse propagation (*Nave and Werner, 2014*). The relevance of myelin for efficient motor, sensory, and cognitive performance is illustrated by their decline in dysmyelinating and demyelinating disorders, including multiple sclerosis (MS) and leukodystrophies and in respective mouse models (*Stadelmann et al., 2019*). MS is a human-specific autoimmune disorder for which it has remained difficult to establish a genuine mouse model, an observation that might point to the existence of human-specific antigens in myelin. Generated by mature oligodendrocytes (MOL), myelin consists of multiple concentric layers of specialized plasma membrane. The ultrastructure of myelin is highly ordered with closely apposed, compacted membrane layers and a non-compacted cytoplasmic channel system that includes the adaxonal myelin layer and paranodal loops. The formation of these subcompartments is enabled by highly enriched, specialized myelin proteins. For example, the transmembrane-tetraspan proteolipid protein (PLP) supports extracellular membrane apposition and adhesion (*Duncan et al., 1987*), the cytoplasmic myelin basic protein (MBP) mediates intracellular membrane apposition (*Popko et al., 1987*), and the enzyme cyclic nucleotide phosphodiesterase (CNP) contributes to structuring noncompact myelin compartments (*Edgar et al., 2009*; *Snaidero et al., 2017*). In fact, using the gel-based methods available at that time, PLP, MBP, and CNP were early recognized as exceptionally abundant myelin proteins in the CNS of tetrapods (*Morell et al., 1973*; *Franz et al., 1981*). Since then, the number of known myelin proteins has markedly increased, including proteins with enzymatic, metabolic, cytoskeletal, adhesive, and immune-related functions (*Nave and Werner, 2014*), and it became possible to quantify their relative abundance by mass spectrometry (*Jahn et al., 2020*).

Myelin biology is primarily studied in mice and zebrafish (*Ackerman and Monk, 2016*). The considerable differences between the species, including dimension and morphology of bodies and brains,

motor performance, cognition, and ecosystem, are owed to evolutionary changes since their last common ancestor about 420 million years ago (mya) (*Ravi and Venkatesh, 2018*). Already early gel-based comparisons between CNS myelin fractions purified from various fish and tetrapod species revealed that the clades comprise overlapping but divergent sets of major myelin proteins (*Franz et al., 1981*; *Yoshida and Colman, 1996*). More recently, this finding was extended to low-abundant constituents when quantitative mass spectrometry allowed comparing the CNS myelin proteome between zebrafish and mice (*Siems et al., 2021*). By quantitative proteome analysis, MBP is highly abundant in CNS myelin of either species. Apart from MBP, however, their myelin proteome differs qualitatively and quantitatively. Thus, the protein composition of myelin displays species-dependent diversity, which can be assessed by mass spectrometry.

No animal model provides an exact replica of the human nervous system. The rodent and primate clades diverged approximately 85 mya (*Yang et al., 2004*). In consequence, mice and humans went through considerable evolutionary time since their last common ancestor. To understand myelination and myelin-related diseases in humans, it is thus relevant to investigate the molecular profiles of human oligodendrocytes and myelin, and, optimally, compare them with their orthologs in relevant model species. There are evident ethical and methodological limitations to studies involving living humans. However, postmortem material donated for scientific assessment has become available. For example, the rates of oligodendrocyte turnover and myelin renewal have been evaluated in humans, including in MS patients (*Yeung et al., 2019*). It has also become possible to determine transcriptional profiles of oligodendrocytes in both humans and mice, including in disease conditions (*Falcão et al., 2018*; *Jäkel et al., 2019*; *Zhou et al., 2020*).

In this study, we used quantitative mass spectrometry to systematically examine the protein composition of myelin purified from the subcortical white matter of human subjects post mortem. Whereas the relative abundance of many structural myelin proteins is roughly similar between human and mouse CNS myelin – the latter was recently established by assessing C56Bl/6N mouse brains using the same methodology (*Jahn et al., 2020*) – we observed striking qualitative and quantitative differences in the relative abundance of multiple other myelin proteins. By integrating and comparing previously established scRNA-seq datasets, we found that their presence in myelin is reflected in the transcriptome profiles of MOL, at least to some extent. Our findings thus reveal unexpected differences in the molecular profiles of CNS myelin and oligodendrocytes between humans and mice. Considering their evolutionary divergence enables a more informed translation from mouse models to humans.

## Results

### Proteome analysis of human CNS myelin

To systematically identify and quantify the protein constituents of human CNS myelin, we biochemically purified a myelin-enriched lightweight membrane fraction from the subcortical normal-appearing white matter of five human subjects post mortem. By electron microscopic assessment of the myelin fraction, constituents other than multilamellar myelin sheaths were largely absent (*Figure 1—figure supplement 1*), confirming that other membrane fractions had been efficiently removed.

We then subjected both the myelin fraction and the corresponding brain homogenate to solubilization using ASB-14 and high urea concentration, automated in-solution tryptic digest by filter-aided sample preparation (FASP), peptide fractionation by nanoUPLC, and ESI-QTOF mass spectrometry involving data-independent acquisition (DIA) of data. The utilized MS$^E$ mode facilitates simultaneous quantification and identification of all peptides entering the mass spectrometer. Proteins can thus be quantified by correlating signal intensities of peptides with those of a spike-in protein of known concentration (TOP3 method; *Silva et al., 2006*). When assessing myelin by MS$^E$, we quantified 332 proteins (*Figure 1—source data 1*; labeled in green in *Figure 1a*) with a false discovery rate (FDR) of <1% and an average sequence coverage of 39.6%. When using the ultra-definition (UD)MS$^E$ mode, in which ion mobility spectrometry enables an additional separation of peptides after chromatography and before mass measurement, we identified and quantified 835 proteins with an average sequence coverage of 37.0% (*Figure 1—source data 1*; labeled in blue in *Figure 1a*). The MS$^E$ mode quantified myelin proteins with a dynamic range of over four orders of magnitude parts per million (ppm), thereby allowing the reliable quantification of all myelin constituents, including the exceptionally abundant

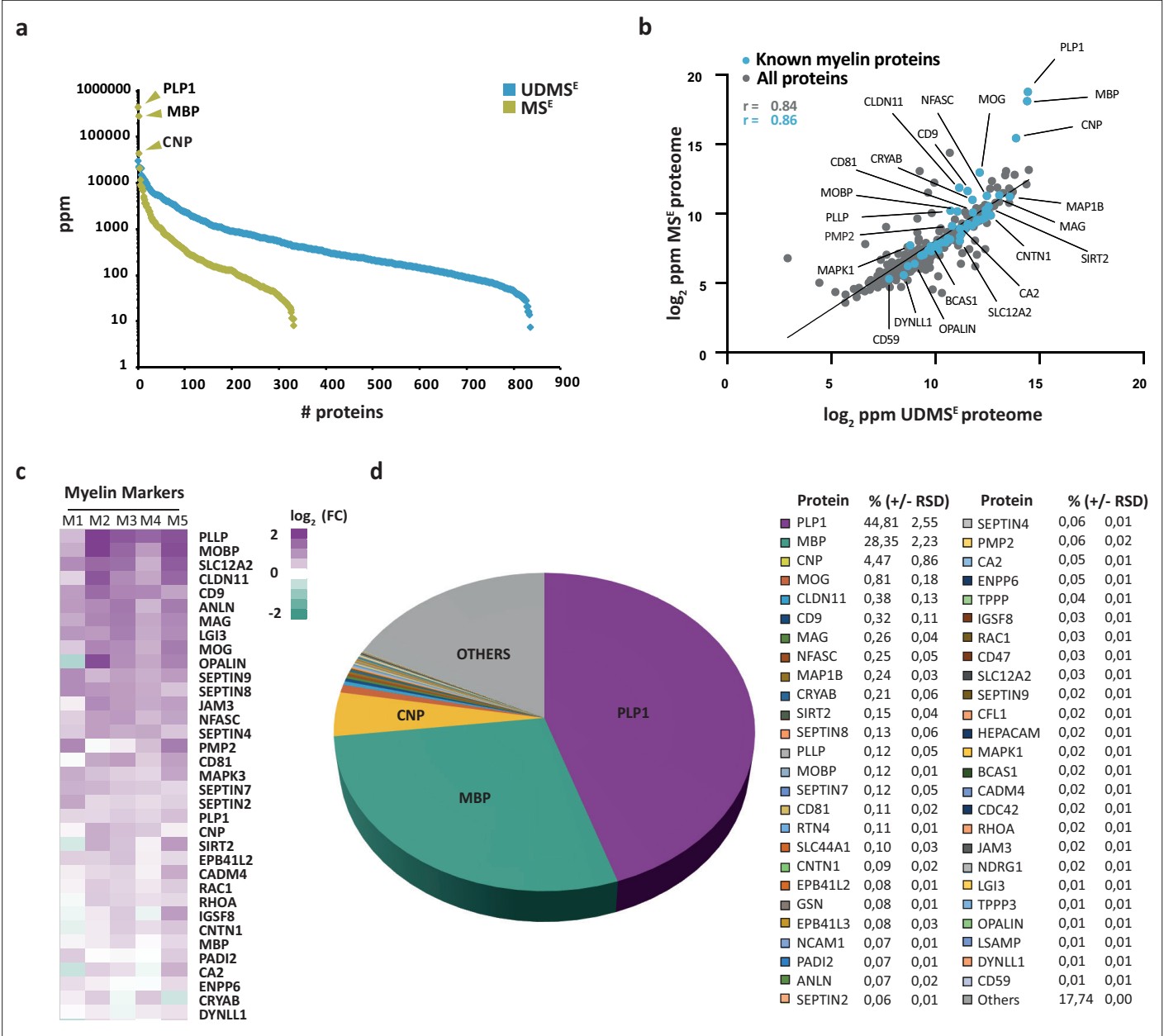

**Figure 1.** Proteome analysis of human central nervous system (CNS) myelin. (**a**) Number and relative abundance of proteins identified in myelin purified from human normal-appearing white matter according to two data-independent acquisition (DIA) mass spectrometric modes (MS$^E$, UDMS$^E$). Note that UDMS$^E$ (blue) identifies a larger number of proteins in myelin but provides a lower dynamic range of quantification. MS$^E$ (lime green) identifies fewer proteins, but the comparatively higher dynamic range of quantification provides information about the relative abundance of the exceptionally abundant myelin proteins proteolipid protein 1 (PLP1), myelin basic protein (MBP), and cyclic nucleotide phosphodiesterase (CNP). See *Figure 1— source data 1* for datasets. ppm, parts per million. (**b**) Scatter plot comparing the log$_2$-transformed relative abundance of proteins identified in myelin by MS$^E$ against their abundance as identified by UDMS$^E$. Data points highlighted in blue represent known myelin proteins, some of which are indicated. The correlation coefficient (r) was calculated for all proteins identified by MS$^E$ (gray) and known myelin proteins (blue). The regression line serves as navigational mean. Note that PLP, MBP, and CNP deviate the most from the regression line due to the limitations of UDMS$^E$ in the correct quantification of such exceptionally abundant myelin proteins. (**c**) Heatmap comparing the relative abundance of known myelin proteins in purified myelin compared to white matter homogenate. Mass spectrometric quantification based on five biological replicates (**M1, M2, M3, M4, M5**) as the average of two technical replicates each. Each horizontal line displays the fold change (FC) of a known myelin protein of which the abundance is increased (magenta) or reduced (turquoise) in human myelin compared to its average abundance in white matter lysate plotted on a log$_2$ color scale. As to the technical quality of the proteomic data and the purity of the myelin fraction, also see Pearson's correlation coefficients in *Figure 1—figure supplement 2* and heatmap comparisons for marker proteins representing other cell types and organelles in *Figure 1—figure supplement 3*. (**d**) Pie chart showing the relative abundance of proteins identified by MS$^E$ in myelin purified from the human white matter. Relative abundance is given in percentage with relative

*Figure 1 continued on next page*

*Figure 1 continued*
standard deviation (% ±RSD). Note that known myelin proteins constitute approximately 82% of the total myelin protein; proteins so far not known as myelin proteins constitute about 18%.

The online version of this article includes the following source data and figure supplement(s) for figure 1:

**Source data 1.** Label-free quantification of proteins in human central nervous system (CNS) myelin and white matter homogenate by two different data acquisition modes.

**Figure supplement 1.** Electron micrograph of the myelin-enriched fraction.

**Figure supplement 2.** Pearson's correlation for proteome analysis by MS$^E$ and UDMS$^E$.

**Figure supplement 3.** Heatmaps comparing the relative abundance of marker proteins in purified myelin versus white matter homogenate.

PLP, MBP, and CNP. The UDMS$^E$ mode identified over twice as many proteins as MS$^E$, though with a compressed dynamic range of only about three orders of magnitude ppm. Expectedly, the MS$^E$ and UDMS$^E$ datasets correlated well with a correlation coefficient of >0.8 (*Figure 1b*, *Figure 1—figure supplement 2*). Both datasets taken together, we identified 848 proteins in human CNS myelin by liquid chromatography–mass spectrometry (LC-MS) analysis. Importantly, the strategy of direct label-free quantification provides information about the relative abundance of identified proteins. When comparing the relative abundance of proteins in the myelin fraction and the corresponding homogenate, we found known myelin markers enriched in the myelin fraction (*Figure 1c*). Markers for other cell types or compartments were either reduced in abundance in the myelin fraction compared to brain lysate or not identified at all (*Figure 1—figure supplement 3*). This indicates that the fraction is suited for proteomic analysis of human myelin.

## Relative abundance of CNS myelin proteins in humans

We used the MS$^E$ dataset to calculate the relative abundance of myelin proteins in the human white matter (*Figure 1d*), considering that quantification of exceptionally abundant proteins requires a high dynamic range. The most abundant myelin proteins were the structural constituents PLP, MBP, and CNP, which accounted for 44.8, 28.4, and 4.5% of the total myelin proteins, respectively. In addition, numerous known myelin proteins were identified and quantified at lower abundance (*Figure 1d*). Previously known myelin proteins constituted approximately 82% of the total human myelin protein (*Figure 1d*), while the remaining 18% were accounted for by other proteins, including occasional contaminants from other cellular sources (*Figure 1—figure supplement 3*).

## Comparison to the mouse myelin proteome

We hypothesized that the protein composition of human and mouse myelin displays some degree of divergence. To compare human and mouse myelin, we first separated myelin of both species by SDS-PAGE. By silver staining, the band patterns were roughly comparable but not identical (*Figure 2a*), supporting the hypothesis that some differences exist. To elucidate differences at the molecular level, we compared the present human mass spectrometric data (ProteomeXchange Consortium PRIDE partner repository, dataset identifier PXD029727) with those of our recent proteomic analysis of myelin purified from the brains of C57Bl/6N mice using the same workflow and methodology (*Jahn et al., 2020*) (dataset identifier PXD020007). As expected, the majority of known myelin proteins were identified in myelin of both species (*Figure 2b*). However, a subset of known myelin proteins was identified only in either human or mouse myelin (*Figure 2b*), in agreement with the hypothesis that the protein composition of myelin is not identical across these species.

For example, we noted that peripheral myelin protein 2 (PMP2, also termed P2 or fatty acid binding protein [FABP8]) was identified in human CNS myelin (*Figure 2b*). PMP2 has long been known as a constituent of myelin in the peripheral nervous system (PNS) synthesized by Schwann cells (*Greenfield et al., 1973*; *Uusitalo et al., 2021*) but based on rodent studies was assumed to be absent from CNS myelin. Yet, PMP2 was readily detected in human CNS myelin by both immunoblotting (*Figure 2—figure supplement 1a*) and immunohistochemistry (*Figure 2—figure supplement 1b*), thus confirming its mass spectrometric identification. In contrast, PMP2 was not detected in mouse CNS myelin by immunoblot (*Figure 2—figure supplement 1a*). In agreement with prior work (*Greenfield et al., 1973*), PMP2 was readily detected in mouse PNS myelin (*Figure 2—figure supplement 1a*), indicating that the utilized antibodies detect PMP2 of either species. Together, this substantiates

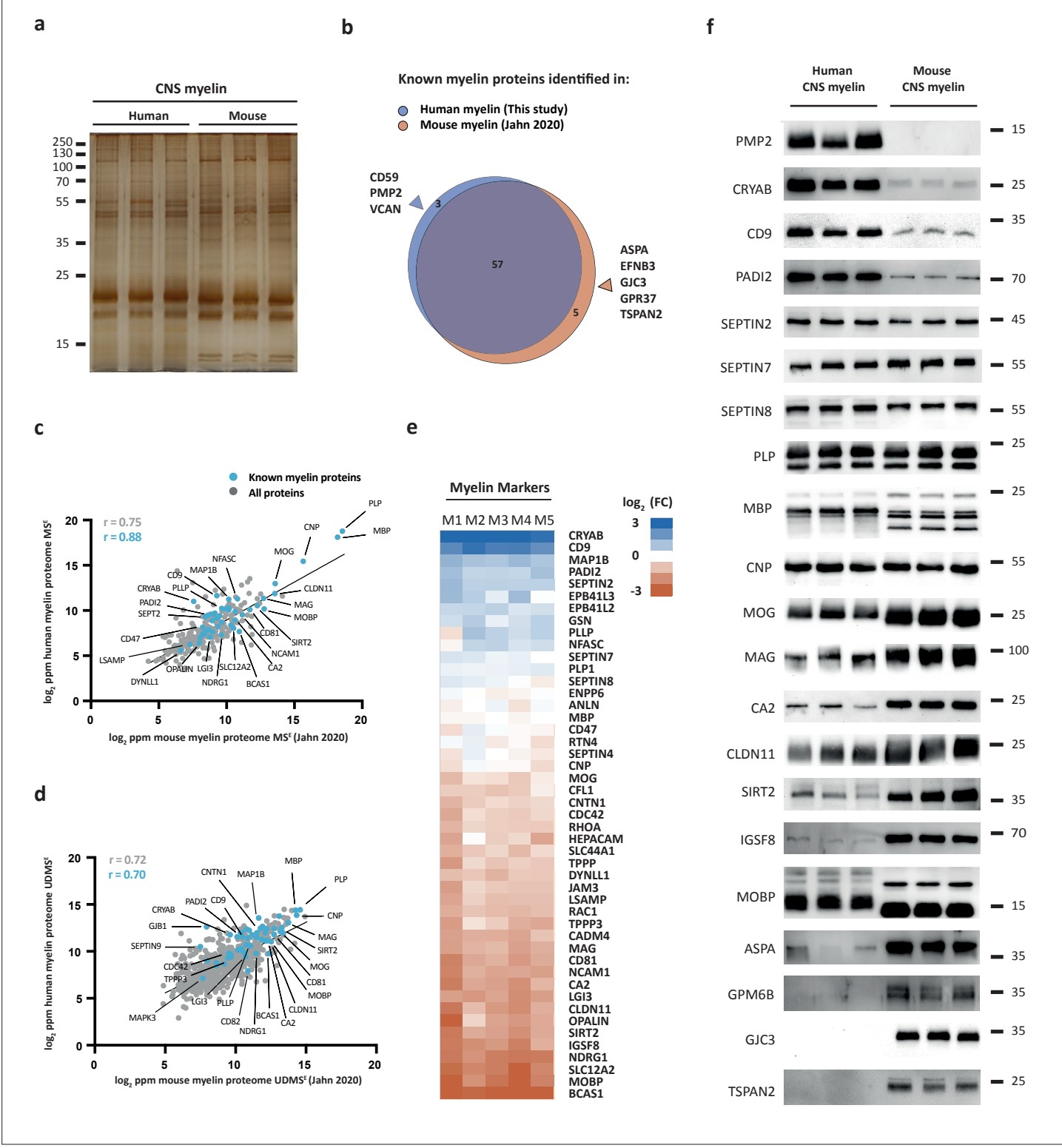

**Figure 2.** Comparison of the protein composition of human and mouse central nervous system (CNS) myelin. (**a**) Silver-stained SDS-PAGE (0.9 µg protein load) of myelin purified from human normal-appearing white matter and C57Bl/6N mouse brains. Note that the band patterns are roughly comparable but not identical. Gel shows n = 3 biological replicates per species. (**b**) Venn diagram comparing 65 selected known myelin proteins identified by MS[E] and UDMS[E] in myelin purified from human white matter (blue) and C57Bl/6N mouse brains (orange) as recently established using the same methods (*Jahn et al., 2020*). Note that most known myelin proteins were identified in myelin of both species, while multiple myelin proteins were identified in myelin of only one species. (**c, d**) Scatter plots of the log$_2$-transformed relative abundance of proteins identified in human myelin

*Figure 2 continued on next page*

*Figure 2 continued*

by MS$^E$ (**c**) or UDMS$^E$ (**d**) plotted against their relative abundance in mouse myelin as recently established using the same methods (*Jahn et al., 2020*). Correlation coefficients (r) were calculated for all proteins identified in human myelin (gray) or known myelin proteins (blue). Regression lines serve as navigational mean. (**e**) Heatmap comparing the relative abundance of known myelin proteins identified by MS$^E$ in human myelin with that in mouse myelin according to the same method (*Jahn et al., 2020*). Each horizontal line displays the fold change (FC) of a protein in five biological replicates (**M1–M5**) of human myelin compared to its average abundance in CNS myelin of mice plotted on a log$_2$-color scale. Note that several proteins display higher abundance in human (blue) or mouse (orange) myelin, while others show approximately similar relative abundance (white). (**f**) Immunoblot analysis confirms comparatively higher abundance in human myelin of PMP2, CRYAB, CD9, and PADI2, approximately equal abundance of PLP, CNP, SEPTIN2, SEPTIN7, and SEPTIN8, and comparatively higher abundance in mouse myelin of TSPAN2, GPM6B, GJC3, ASPA, MOBP, IGSF8, SIRT2, CLDN11, CA2, MAG, and MOG, as implied by the MS$^E$ analysis. Note that immunoblot-based comparison of the relative abundance of MBP across species is not straightforward because MBP displays one dominant isoform (18.5 kDa) in human CNS myelin but three main isoforms (14.0, 17.0, and 18.5 kDa) in mouse CNS myelin due to species-dependent alternative splicing. Blots show n = 3 biological replicates per species. For immunohistochemistry detecting PMP2 in human optic nerve cross sections, see *Figure 2—figure supplement 1*.

The online version of this article includes the following source data and figure supplement(s) for figure 2:

**Source data 1.** Labeled original immunoblots.

**Figure supplement 1.** Detection of PMP2 in human central nervous system (CNS) myelin by immunoblot and immunohistochemistry.

the existence of species-dependent differences in the protein composition of CNS myelin between humans and mice.

Next, we plotted all proteins identified in human CNS myelin, that is, the present MS$^E$ and UDMS$^E$ datasets, against those identified in mouse myelin as recently established using the same methodology (*Jahn et al., 2020*) (dataset identifier PXD020007). Indeed, the datasets correlated well with correlation coefficients of >0.7 (MS$^E$, *Figure 2c*; UDMS$^E$, *Figure 2d*) but clearly diverged to some extent. We therefore cross-compared the abundance of individual myelin proteins in human and mouse myelin by MS$^E$ using heatmap visualization (*Figure 2e*). We found that major structural myelin proteins, including PLP, MBP, CNP, SEPTIN2, SEPTIN7, and SEPTIN8, displayed a similar relative abundance in myelin of both species. However, several other myelin proteins were comparatively more abundant in human myelin, as exemplified by crystallin-αB (CRYAB), CD9 (also termed tetraspanin-29 [TSPAN29]), and peptidyl arginine deiminase (PADI2), or in mouse myelin, including myelin-associated oligodendrocyte basic protein (MOBP), sirtuin-2 (SIRT2), and carbonic anhydrase 2 (CA2). Importantly, when detecting these proteins by immunoblotting in myelin of both species (*Figure 2f*), these results were generally consistent with the mass spectrometric comparison (*Figure 2e*). Yet, quantitative mass spectrometry emerged as more straightforward than immunoblotting when comparing the relative abundance of proteins across species if species-dependent differences in splice isoforms exist. This is exemplified by MBP, which – owing to species-dependent alternative splicing (*Campagnoni, 1988*) – displays three main isoforms (14.0, 17.0, and 18.5 kDa) in mouse CNS myelin but only one dominant isoform (18.5 kDa) in human CNS myelin, in agreement with previous observations (*Waehneldt and Malotka, 1980*; *Ishii et al., 2009*). Taken together, the protein composition of human and mouse CNS myelin is similar with respect to the relative abundance of major structural proteins but displays remarkable qualitative and quantitative differences regarding many other myelin proteins.

## Integrated scRNA-seq profile of human and mouse MOL

To identify species-dependent transcriptional differences that may underlie the diversity of the myelin proteome, we utilized high-resolution mRNA-abundance profiles to assess the oligodendrocyte lineage in both humans and mice. To this aim, we retrieved previously published scRNA-seq datasets from the CNS of humans (*Jäkel et al., 2019*; *Zhou et al., 2020*; *Lake et al., 2018*; *Habib et al., 2017*; *Grubman et al., 2019*; *Wheeler et al., 2020*) and mice (*Falcão et al., 2018*; *Wheeler et al., 2020*; *Marques et al., 2016*; *Saunders et al., 2018*; *Zeisel et al., 2018*; *Zeisel et al., 2015*; *Ximerakis et al., 2019*) and evaluated all cells designated as oligodendrocyte progenitor cells (OPCs), newly formed oligodendrocytes (NFOs), and MOL from nondiseased subjects (*Figure 3—figure supplement 1a and b*, *Figure 3—source data 1*). Using the SCTransform pipeline within the R toolkit Seurat, it was possible to integrate cells from all available human and mouse datasets into respective single objects (*Figure 3—figure supplement 1a and b*). Importantly, cells from all studies distributed well across the uniform manifold approximation and projection (UMAP) plots (*Figure 3—figure supplement 1a and b*), implying suitability for integration and further assessment. Indeed, when highlighting

marker gene expression on UMAPs, cells expressing markers for OPCs (*CSPG4, PCDH15, PDGFRA, PTPRZ1*) or MOL (*ANLN, CNP, MBP, PLP1*) clustered well in both the human and mouse integrated datasets (*Figure 3—figure supplement 1c and e*). Notably, however, multiple myelin-related transcripts displayed considerable expression only in human or mouse oligodendrocytes, as exemplified by *TSPAN2, GJC3,* and *PMP2* (*Figure 3—figure supplement 1f*). We noted that cells expressing established NFO markers (*BMP4, ENPP6, FYN, GPR17*) clustered well in the mouse but not the human integrated dataset (*Figure 3—figure supplement 1d*), probably owing to the low number of NFO in the latter. Indeed, only 132 cells designated as NFO were comprised in the available human scRNA-seq datasets, considerably fewer compared to 10,391 NFO recovered from the mouse datasets (*Figure 3—source data 1*). Considering that the number of cells designated as NFO in the human datasets is probably too low for a reasonable bioinformatic comparison, we focused on MOL for a more thorough species-dependent comparison of transcriptional profiles of myelin-related genes.

To this aim, we subset all cells annotated as MOL from control samples in all datasets of both species (*Falcão et al., 2018*; *Jäkel et al., 2019*; *Zhou et al., 2020*; *Lake et al., 2018*; *Habib et al., 2017*; *Grubman et al., 2019*; *Wheeler et al., 2020*; *Marques et al., 2016*; *Saunders et al., 2018*; *Zeisel et al., 2018*; *Zeisel et al., 2015*; *Ximerakis et al., 2019*; *Figure 3a*) for integration via SCTransform. Mouse gene symbols were translated to human gene symbols prior to data integration. Importantly, cells from both species distributed well across the UMAPs (*Figure 3b and c*), providing the basis for assessing the transcriptional profiles of 41,517 human and 95,966 mouse MOL. At the level of gene expression, cells expressing myelin marker transcripts (*ANLN, CNP, GSN, MBP, PLLP, PLP1*) distributed similarly across human and mouse MOL (*Figure 3d*), as did transcripts encoding myelination-related transcription factors (*MYRF, SOX10*) (*Figure 3e*). Notably, multiple myelin-related transcripts displayed exclusive or predominant expression in MOL of only one of the species, as exemplified by *PMP2, PADI2, CA2, TSPAN2,* and *GJC3* (*Figure 3f*). Also, when assessed in the species-separately integrated datasets of all oligodendroglial cells, including OPC, NFO, and MOL, these genes displayed little or no expression in the respective other species (*Figure 3—figure supplement 1f*).

To compare the transcriptome profiles between human and mouse MOL without the influence of sequencing batch effects, we first applied van der Waerden (vdW)-ordered quantile transformation to the mean mRNA abundance values of 3000 integration features (i.e., genes) in all cells designated as MOL in the utilized datasets. We then plotted the average relative transcript abundance in human versus mouse MOL (*Figure 3—figure supplement 2*). When comparing all 3000 integration features, the averaged vdW-normalized mRNA-abundance profiles correlated reasonably well with a coefficient of 0.59 (gray data points in *Figure 3—figure supplement 2*). However, a comparatively higher correlation coefficient of 0.84 was found when comparing only known myelin-related transcripts (n = 37 transcripts; highlighted as blue data points in *Figure 3—figure supplement 2*). This implies that the transcriptional profiles of known myelin-related transcripts are more similar between humans and mice than those of the other transcripts expressed in MOL. In particular, the abundance of several transcripts encoding structural myelin proteins, including *PLP1, MBP,* and *CNP,* was essentially equal between human and mouse MOL. In similarity, the abundance of transcripts for myelin-related transcription factors (*MYRF, OLIG1, OLIG2, SOX10*) was also roughly similar (labeled in green in *Figure 3—figure supplement 2*). We noted that the most abundant transcripts in MOL also included genes of which the protein products were not mass spectrometrically identified in myelin – and thus not comprised in the myelin proteome. As an example, the abundance of the *Mal*-transcript encoding myelin and lymphocyte protein (MAL) correlates well between human and mouse MOL (labeled in orange *Figure 3—figure supplement 2*). MAL is a known myelin protein (*Schaeren-Wiemers et al., 2004*), which, however, is not identified by mass spectrometry because of its nonsuitable tryptic digest pattern. Taken together, the integrated scRNA-seq profiles of MOL generally correlated well between humans and mice, particularly with respect to known myelin-related mRNAs. However, several transcripts displayed a qualitatively or quantitatively divergent abundance when compared between the species (*Figure 3f*, *Figure 3—figure supplement 1f*, *Figure 3—figure supplement 2*).

Next, we compared our human (*Figure 1—source data 1*) and mouse myelin (*Jahn et al., 2020*) proteome datasets (by $MS^E$ or $UDMS^E$) with the averaged vdW-normalized mRNA abundance profile of MOL in the respective same species. We calculated correlation coefficients of <0.5 (*Figure 3—figure supplement 3*). The degree of correlation between the myelin proteome and the MOL transcriptome within the same species is thus considerably lower than that between the myelin proteome

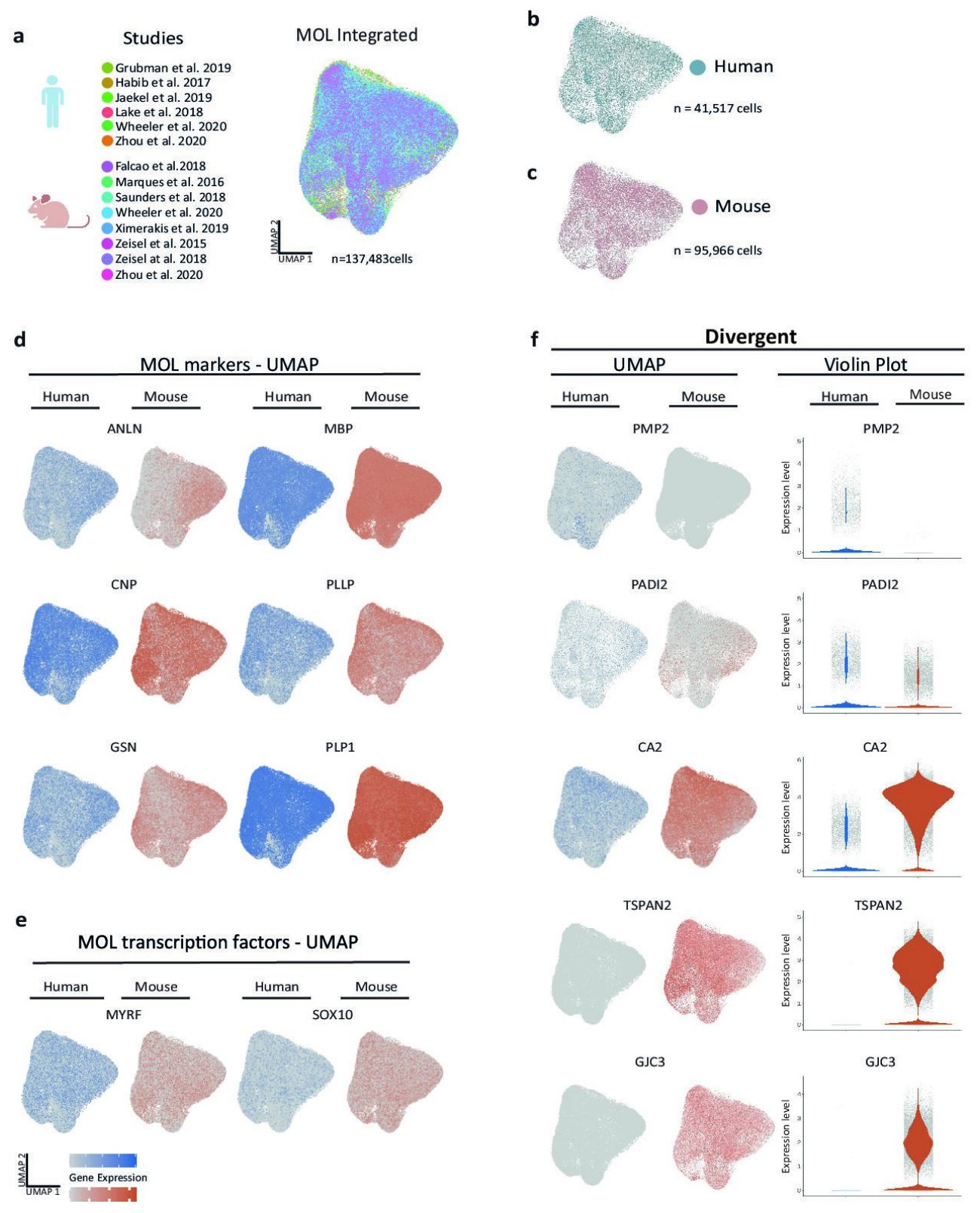

**Figure 3.** Cross-species scRNA-seq profile comparison of mature oligodendrocytes (MOL). (**a–c**) Uniform manifold approximation and projection (UMAP) plot of the scRNA-seq profile of MOL integrated from previously established human (**b**) and mouse (**c**) datasets. In (**a**), cells contributed by distinct studies are highlighted in different colors; the corresponding references are given. (**d, e**) UMAP feature plots highlighting expression of selected

*Figure 3 continued on next page*

*Figure 3 continued*

MOL marker genes (**d**) and transcription factors (**e**) in the integrated object comprising MOL of both humans (blue) and mice (orange). (**f**) UMAP feature plots and violin plots exemplify genes that display preferential expression in MOL of humans (*PMP2, PADI2*) or mice (*GJC3, TSPAN2, CA2*).

The online version of this article includes the following source data and figure supplement(s) for figure 3:

**Source data 1.** Parameters applied for scRNA-seq individual dataset quality control and integrative analysis.

**Figure supplement 1.** Integrated scRNA-seq profiles of the oligodendrocyte lineage in humans and mice.

**Figure supplement 2.** Cross-species mature oligodendrocyte (MOL) transcriptome correlation.

**Figure supplement 3.** Comparisons of myelin proteome and mature oligodendrocyte (MOL) transcriptome.

---

(*Figure 2c and d*) or the MOL transcriptome (*Figure 3—figure supplement 2*), respectively, across the two species.

## Subpopulation analysis of integrated human and mouse MOL scRNA-seq profile

Previously, multiple transcriptome studies have identified distinct subpopulations of MOL in both humans and mice (*Falcão et al., 2018*; *Jäkel et al., 2019*; *Marques et al., 2016*), which were correlated according to the expression of marker genes. Here, we tested whether similar subpopulations also manifest if evaluating the merged and integrated scRNAseq profiles, thereby not only allowing cross-species comparison but also increasing the dimensionality of assessed MOL per species. Indeed, k-nearest neighbor (KNN) clustering identified five potential subpopulations of MOL (labeled as clusters 0, 1, 2, 3, and 4 in *Figure 4a–c*). Notably, all subpopulations displayed approximately similar expression levels of marker genes encoding structural myelin proteins (*PLP1, MBP, CNP, CLDN11, MAG*) (*Figure 4b*). However, the subpopulations were defined by varying degrees of expression of other transcripts, including the myelin-related *CD9* and *OPALIN* (cluster 0), *APOD, KLK6,* and *S100B* (cluster 1), *APOE* and *CST3* (cluster 2), *CA2* and *PTGS* (cluster 3), and *SIRT2* and *NFASC* (cluster 4) (*Figure 4b*). Considering the larger number of evaluated cells compared to the prior individual studies on which the present assessment is based, these findings support the previously identified subpopulations of MOL (*Jäkel et al., 2019*; *Marques et al., 2016*). Based on Gene Ontology (GO) term enrichment analysis of biological processes (*Figure 4—figure supplement 1*), one may speculate that MOL in clusters 2–4 are associated with GO terms grouped as protein synthesis, electron transport, and immune activation, respectively. However, their functional specialization and relevance remain to be shown. Less speculatively, both human and mouse MOL comprise all five subpopulations to an approximately similar extent (*Figure 4c*), implying that none of these MOL subpopulations is restricted to either one of these species.

## Discussion

We performed quantitative proteome analysis to determine the protein composition of human CNS myelin. Subjecting myelin biochemically purified from human subcortical white matter to label-free mass spectrometry allowed identifying hundreds of proteins with very high confidence. More importantly, the method involves quantifying peptide intensities without prefractionation, thereby providing direct information about the relative abundance of myelin proteins. The latter provides a considerable advancement compared to previous approaches involving prefractionation at the protein level via 1D gels (*Ishii et al., 2009*; *Martins-De-Souza, 2020*) or at the peptide level via 2D liquid chromatography (*Dhaunchak et al., 2010*; *Gopalakrishnan et al., 2013*), which yielded lists of proteins identified in human CNS myelin but without information about their relative abundance.

Knowing the relative abundance of myelin proteins enables the assessment of both their stoichiometric relationships and cross-species comparisons. For example, the filament-forming septins SEPTIN2, SEPTIN4, SEPTIN7, and SEPTIN8 displayed a molar stoichiometry of about 1:1:2:2 in human CNS myelin. Notably, the same septin subunits are also comprised in myelin of mice with a similar molar stoichiometry (*Jahn et al., 2020*), a likely prerequisite for their assembly into similar core multimers and higher-order structures. Indeed, experiments in mice have previously shown that these septin subunits assemble into membrane-associated filaments that stabilize the adaxonal compartment of CNS myelin (*Patzig et al., 2016*). Integrating the current view on septin assembly (*Soroor*

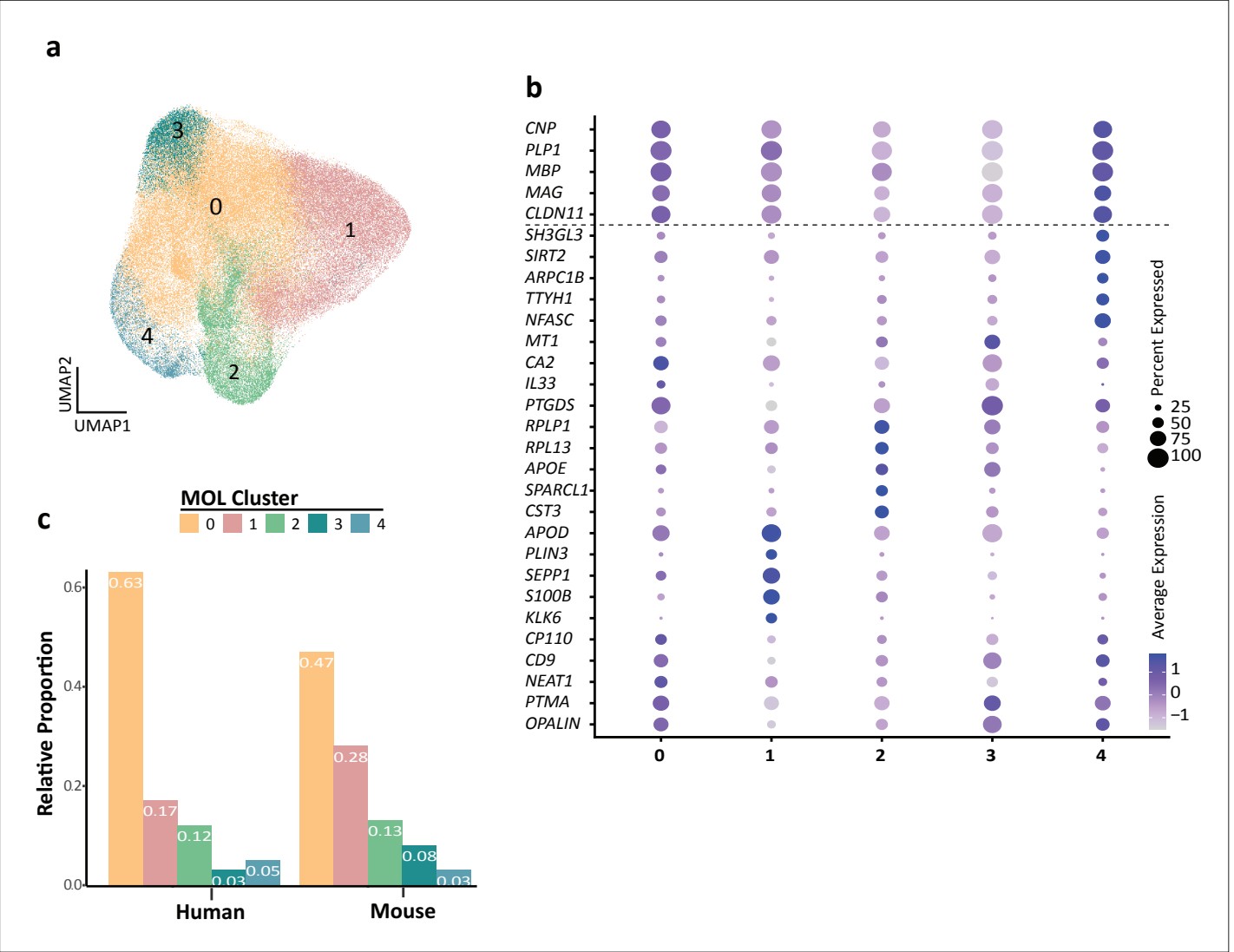

**Figure 4.** Human and mouse mature oligodendrocyte (MOL) subpopulation analysis. (**a**) Uniform manifold approximation and projection (UMAP) plot showing five subpopulations of MOL identified upon integrating all human and mouse scRNA-seq datasets. (**b**) Bubble plot showing the top five subpopulation marker genes. All cells in all clusters also express high levels of known myelin-related marker transcripts (*CNP, PLP1, MBP, MAG, CLDN11*). (**c**) Relative proportion of mature oligodendrocyte subpopulations in humans and mice. Note that the MOLs of both species contribute to all subpopulations.

The online version of this article includes the following source data and figure supplement(s) for figure 4:

**Source data 1.** Model-based analysis of single-cell transcriptomics (MAST)-calculated marker genes from human and mouse integrated mature oligodendrocyte (MOL) subpopulations.

**Figure supplement 1.** Gene Ontology (GO) term topics enriched per subpopulation.

*et al., 2021*) and the relative abundance of septin subunits in myelin, it is possible to deduce that the predominant core multimer in myelin is a hexamer of septins 2/4-8-7-7-8-2/4. The comparatively low abundance of SEPTIN9 in myelin implies that core octamers occur less frequently. Assessing the relative abundance of myelin proteins also allows deducing that for each core hexamer up to one molecule of the adaptor protein anillin that facilitates septin assembly (*Erwig et al., 2019b*) is present in CNS myelin. Together, the relative abundance and multimer composition of myelin septins emerge as conserved between human and mouse CNS myelin, similar to that of other structural myelin proteins, including PLP, MBP and CNP.

On the other hand, we also found considerable qualitative and quantitative differences between the protein composition of human and mouse CNS myelin. For example, the tetraspan-transmembrane

proteins TSPAN2 (*Birling et al., 1999*; *Terada et al., 2002*; *de Monasterio-Schrader et al., 2013*) and GPM6B/M6B/Rhombex29 (*Shimokawa and Miura, 2000*; *Werner et al., 2013*) were previously established as myelin proteins in mice and rats, and they were readily identified in CNS myelin of mice by both mass spectrometry (*Jahn et al., 2020*) and immunoblot. However, these proteins were of very low abundance or virtually undetectable in human myelin by both techniques. It has been established in experimental mice that TSPAN2 and GPM6B contribute to immunomodulation (*de Monasterio-Schrader et al., 2013*) and myelin biogenesis (*Werner et al., 2013*), respectively. The present data thus imply that the mouse and human orthologs do not contribute equally to these functions. The gap junction protein GJC3/CX29 has also been established as a myelin protein in mice and rats and is thought to mediate intercellular coupling via gap junctions (*Nagy et al., 2003*; *Kleopa et al., 2004*). Similar to TSPAN2 and GPM6B, GJC3 has been readily identified mass spectrometrically and by immunoblotting in mouse but not human myelin. Interestingly, though, deletion of the *Gjc3*-gene in mice did not have evident morphological or functional consequences for CNS myelin or oligodendrocytes (*Altevogt and Paul, 2004*; *Eiberger et al., 2006*). The benefit for mouse CNS myelin to comprise GJC3 thus remains unknown at this time.

On the other hand, PMP2 (previously termed P2 or FABP8), a membrane-phosphoinositide-binding protein (*Abe et al., 2021*), has long been known as a constituent of peripheral myelin generated by Schwann cells in the PNS (*Uusitalo et al., 2021*; *Brostoff et al., 1975*; *Kitamura et al., 1980*) and actually considered a marker to discriminate peripheral from central myelin (*Franz et al., 1981*; *DeArmond et al., 1980*; *Whitaker, 1981*). Notably, the experiments establishing this view involved bovine and rodent but not human samples. Our finding that PMP2 is a myelin protein in the human CNS leaves open the question of whether this reflects clade-specific de novo recruitment into CNS myelin or selective constraints that eliminated PMP2 from myelin in the clade including cows and rodents. At the evolutionary level, it is interesting to speculate which benefits (if any) human myelin may have from comprising PMP2 or what the evolutionary constraints may be that led to its dropout from rodent CNS myelin. A lead may come from the investigation of the PNS of *Pmp2*-deficient mice, which displayed an altered myelin lipid profile associated with reduced motor nerve conduction velocity (*Zenker et al., 2014*). It is tempting to speculate that the presence of PMP2 in human but not mouse CNS myelin may affect the composition or organization of its lipids, and, possibly, conduction velocity. The *PMP2* gene causes – when mutated – the peripheral neuropathy Charcot–Marie–Tooth (CMT) disease type 1G (*Motley et al., 2016*; *Hong et al., 2016*). A subset of these patients has been tested by brain MRI; however, no major pathology of the white matter was found that would be typical of a leukodystrophy (*Motley et al., 2016*). Yet, our finding that PMP2 is a myelin protein in the human CNS indicates that further testing these and other CMT1G patients for central involvement may find yet-overlooked impairments, possibly more subtle than visible by MRI.

How could species-dependent differences in myelin protein composition come about at the molecular level? Considering the limitations imposed by the availability of human samples, we cannot formally rule out that differences in the sex or age of specimen, brain region, sample preparation, or data analysis may affect the degree of correlation. However, we note that both male and female donors are represented in the human samples, and that instrumentation, methodology, and data analysis were the same in establishing the mouse myelin proteome (*Jahn et al., 2020*) and the human myelin proteome assessed here. The postmortem delay unavoidable for sampling human specimen is unlikely to affect the present comparison when considering that the average postmortem delay is 6 hr at the Netherlands Brain Bank that supplied the human samples used here and that an experimental postmortem delay of 6 hr did not considerably affect the myelin proteome in C57Bl/6N mice (*Jahn et al., 2020*). Finally, we believe that the high degree of cross-species similarity regarding the abundance of structural myelin proteins between humans and mice allows trust in the overall comparison of myelin protein composition, including for proteins displaying cross-species dissimilarity. Thus, individual myelin proteins displaying species-dependent differences may be owing to species-dependent differences in intracellular trafficking and incorporation into the myelin sheath, stability and turnover rate, mRNA-to-protein translation efficiency, or actual mRNA expression.

Indeed, our cross-species integration and comparison of the scRNAseq profiles of MOL imply that species-dependent mRNA expression can explain the species-dependent differences in myelin protein composition at least to some extent. For example, *PMP2/Pmp2* mRNA is expressed in human but not mouse MOL, and PMP2 protein was identified in human but not mouse myelin. Vice versa,

*TSPAN2/Tspan2* and *GJC3/Gjc3* transcripts are expressed in mouse but not human MOL and their protein products TSPAN2 and GJC3 are identified in mouse but not human myelin. Less exclusively, *CA2/Car2* mRNA is preferentially expressed in mouse compared to human MOL, correlating with the relative abundance of its protein product CAR2/CA2 in mouse compared to human myelin. Vice versa, the higher abundance of PADI2 in human compared to mouse myelin goes along with a higher abundance of *PADI2/Padi2* mRNA in human compared to mouse MOL. Together, species-dependent protein abundance in myelin is probably owing to species-dependent mRNA expression, at least for some myelin constituents. We speculate that the evolutionary emergence of regulatory elements that regulate oligodendroglial gene expression in the hominin clade (*Castelijns et al., 2020*) partly underlies speciation of oligodendroglial transcript profiles and myelin protein composition. However, evolutionary changes may also affect oligodendroglial gene regulation specifically in the rodent clade.

It is of note that not all myelin constituents display an evident cross-species correlation between the abundance of their transcripts in myelinating oligodendrocytes and the abundance of the protein products in myelin. Examples include the tetraspanin CD9/TSPAN29 (*Terada et al., 2002*; *Kagawa et al., 1997*), which is more abundant in human compared to mouse myelin, and the enzyme aspartoacylase (ASPA) (*Madhavarao et al., 2004*), which is more abundant in mouse compared to human myelin. Indeed, the abundance of the transcripts encoding either protein is approximately similar when comparing human and mouse MOL. This indicates that not all differences in the protein composition of human and mouse myelin are caused by species-dependent differences in gene expression by MOL. Thus, the efficiency of mRNA translation, intracellular trafficking, incorporation into myelin, and stability or turnover rate of myelin proteins may also display species-dependent efficiency.

In conclusion, both oligodendroglial mRNA abundance profiles and the CNS myelin proteome display widespread similarities between humans and mice, suggesting considerable evolutionary conservation. However, distinct molecular differences were evident, indicating evolutionary recruitment or dropout of myelin proteins across mammalian clades. Mice are commonly assessed as a model for humans in myelin biology. Considering the evolutionary heterogeneity of oligodendroglial mRNA expression and myelin composition can be instructive when translating between mouse models and humans.

## Materials and methods

### Human samples

Postmortem brain tissue was provided by the Netherlands Brain Bank. Donors gave informed consent to perform autopsy and for the use of clinical and pathological information by researchers, approved by the medical ethics committee of the VU Medical Center (Amsterdam, The Netherlands) decided by the Tissue Advisory Board with project number 1191. The diagnoses were confirmed by a neuropathologist.

The following subjects were used for myelin purification and proteome analysis of lysate (L) and purified myelin (M) as well as for immunoblotting:

| Subject | Sex | Age (years) | Diagnosis | PMD (hr) | Brain region | Sample name in mass spectrometry |
|---|---|---|---|---|---|---|
| 1995-106 | Male | 74 | Non-demented control | 08:00 | Subcortical white matter | Control_L1/2 Control_M1/2 |
| 1996-052 | Male | 73 | Non-demented control | 09:10 | Subcortical white matter | Control_L3/4 Control_M3/4 |
| 2002-024 | Female | 75 | Non-demented control | 05:30 | Subcortical white matter | Control_L5/6 Control_M5/6 |
| 2010-015 | Female | 73 | Non-demented control | 07:45 | Subcortical white matter | Control_L7/8 Control_M7/8 |
| 2017-124 | Female | 55 | Non-demented control | 07:30 | Subcortical white matter | Control_L9/10 Control_M9/10 |

The following subjects were used for immunohistochemical analysis:

| Subject | Sex | Age (years) | Diagnosis | PMD (hr) | Brain region | Comments |
|---------|-----|-------------|-----------|----------|--------------|----------|
| 2019-026 | Male | 55 | Parkinson Disease | 05:55 | Optic nerve | No pathology of the optic nerve |
| 2019-077 | Female | 91 | Non-demented control | 09:30 | Optic nerve | Immunolabeled section shown in *Figure 2—figure supplement 1* originates from this subject |
| 2019-106 | Female | 80 | Non-demented control | 06:50 | Optic nerve | - |

## Animal welfare

For the procedure of sacrificing vertebrates for preparation of tissue, all regulations given in the German animal welfare law (TierSchG §4) are followed. Since sacrificing of vertebrates is not an experiment on animals according to §7 Abs. 2 Satz 3 TierSchG, no specific ethical review and approval or notification is required for this work. All procedures were supervised by the animal welfare officer and the animal welfare committee for the Max Planck Institute for Multidisciplinary Sciences, Göttingen, Germany. The animal facility at the Max Planck Institute for Multidisciplinary Sciences is registered according to §11 Abs. 1 TierSchG.

## Myelin purification

A lightweight membrane fraction enriched for myelin was purified from pieces of normal-appearing white matter of human subjects post mortem as specified above, brains of C57Bl/6N mice, and sciatic nerves of C57Bl/6N mice using an established protocol involving two steps of sucrose density gradient centrifugation and osmotic shocks (*Erwig et al., 2019a*). Myelin accumulates at the interface between 0.32 M and 0.85 M sucrose.

## Electron microscopy of purified myelin

For assessment of the human myelin fraction by electron microscopy, myelin purified from the white matter of subjects 1995-106 and 1996-052 was used. 75 µl of each myelin sample was mixed with 75 µl 2× concentrated fixative composed of 5% glutaraldehyde, 8% formaldehyde, and 1.0% NaCl in 100 mM phosphate buffer pH 7.3. Then, the fixed fraction was spun down and resuspended in 2% agarose Super LM (Roth, Karlsruhe, Germany). After solidification, the pellet was cut into two halves and embedded in Epon after postfixation in 2% $OsO_4$. Ultrathin sections across the pellet were prepared using a UC7 ultramicrotome (Leica Microsystems, Vienna, Austria) equipped with a 35° diamond knife (Diatome, Biel, Switzerland). Images were taken with a LEO912 transmission electron microscope (Carl Zeiss Microscopy, Oberkochen, Germany) using a 2k on-axis CCD camera (TRS, Moorenweis, Germany).

## Label-free quantification of myelin proteins

In-solution digestion of myelin proteins according to an automated FASP protocol (*Erwig et al., 2019a*) and LC-MS-analysis by different $MS^E$-type DIA mass spectrometry approaches was performed as recently established for mouse PNS (*Siems et al., 2020*) and CNS (*Jahn et al., 2020*) myelin. Briefly, protein fractions corresponding to 10 µg myelin protein were dissolved in lysis buffer (1% ASB-14, 7 M urea, 2 M thiourea, 10 mM DTT, 0.1 M Tris pH 8.5) and processed according to a CHAPS-based FASP protocol in centrifugal filter units (30 kDa MWCO, Merck Millipore). After removal of the detergents, protein alkylation with iodoacetamide, and buffer exchange to digestion buffer (50 mM ammonium bicarbonate [ABC], 10% acetonitrile), proteins were digested overnight at 37°C with 400 ng trypsin. Tryptic peptides were recovered by centrifugation and extracted with 40 µl of 50 mM ABC and 40 µl of 1% trifluoroacetic acid (TFA), respectively. Combined flow-throughs were directly subjected to LC-MS analysis. For quantification according to the TOP3 approach (*Silva et al., 2006*), aliquots were spiked with 10 fmol/µl of Hi3 EColi standard (Waters Corporation), containing a set of quantified synthetic peptides derived from the *Escherichia coli* chaperone protein ClpB.

Nanoscale reversed-phase UPLC separation of tryptic peptides was performed with a nanoAcquity UPLC system equipped with a Symmetry C18 5 µm, 180 µm × 20 mm trap column and an HSS T3 C18 1.8 µm, 75 µm × 250 mm analytical column (Waters Corporation) maintained at 45°C. Peptides

were separated over 120 min at a flow rate of 300 nl/min with a gradient comprising two linear steps of 3–35% mobile phase B (acetonitrile containing 0.1% formic acid) in 105 min and 35–60% mobile phase B in 15 min, respectively. Mass spectrometric analysis on a quadrupole time-of-flight mass spectrometer with ion mobility option (Synapt G2-S, Waters Corporation) was performed in the ion mobility-enhanced DIA mode with drift time-specific collision energies referred to as UDMS[E] (*Distler et al., 2014*). As established previously for proteome analysis of purified mouse myelin (*Jahn et al., 2020*; *Siems et al., 2020*), samples were rerun in a data acquisition mode without ion mobility separation of peptides (referred to as MS[E]) to ensure the correct quantification of exceptionally abundant myelin proteins. Continuum LC-MS data were processed using Waters ProteinLynx Global Server (PLGS) and searched against a custom database compiled by adding the sequence information for *E. coli* chaperone protein ClpB and porcine trypsin to the UniProtKB/SwissProt human proteome (release 2019-10, 20,379 entries) and by appending the reversed sequence of each entry to enable the determination of FDR. Precursor and fragment ion mass tolerances were automatically determined by PLGS and were typically below 5 ppm for precursor ions and below 10 ppm (root mean square) for fragment ions. Carbamidomethylation of cysteine was specified as fixed and oxidation of methionine as variable modification. One missed trypsin cleavage was allowed. Minimal ion matching requirements were two fragments per peptide, five fragments per protein, and one peptide per protein. FDR for protein identification was set to 1% threshold.

For post-identification analysis, including TOP3 quantification of proteins, ISOQuant (*Distler et al., 2014*; software freely available at www.isoquant.net/) was used as described previously (*Jahn et al., 2020*; *Siems et al., 2020*). Only proteins represented by at least two peptides (minimum length six amino acids, score ≥5.5, identified in at least two runs) were quantified as ppm, that is, the relative amount (w/w) of each protein with respect to the sum over all detected proteins. FDR for both peptides and proteins was set to 1% threshold and at least one unique peptide was required. Human myelin fractions and the corresponding white matter homogenates were assessed as five biological replicates (n = 5) each. The proteome analysis was repeated as an independent replicate experiment from the same protein fractions, resulting in 10 LC-MS runs per condition. The mass spectrometry proteomics data have been deposited to the ProteomeXchange Consortium via the PRIDE (*Perez-Riverol et al., 2019*) partner repository with dataset identifier PXD029727.

## Visualization of proteomic data

Proteomic data were visualized and analyzed as in *Jahn et al., 2020* and *Siems et al., 2020*. In more detail, heatmaps and scatter plots were prepared in Microsoft Excel 2016 and GraphPad Prism 9. The area-proportional Venn diagram was prepared using BioVenn (*Hulsen et al., 2008*).

## Gel electrophoresis and silver staining of gels

Protein concentrations were determined using the DC Protein Assay kit (Bio-Rad, Hercules, CA). Samples were diluted in 1× SDS sample buffer with dithiothreitol and separated on a 12% SDS-PAGE for 1 hr at 200 V using the Bio-Rad system; gels were fixated overnight in 10% (v/v) acetic acid/40% (v/v) ethanol, and then washed in 30% ethanol (2 × 20 min) and ddH$_2$O (1 × 20 min). For sensitization, gels were incubated 1 min in 0.012% (v/v) Na$_2$S$_2$O$_3$ and subsequently washed with ddH$_2$O (3 × 20 s). For silver staining, gels were impregnated for 20 min in 0.2% (w/v) AgNO$_3$/0.04% formaldehyde, washed with ddH$_2$O (3 × 20 s), and developed in 3% (w/v) Na$_2$CO$_3$/0.02% (w/v) formaldehyde. The reaction was stopped by exchanging the solution with 5% (v/v) acetic acid. Gels were kept in ddH$_2$O until documentation.

## Immunoblotting

Immunoblotting was performed as described (*Patzig et al., 2016*). Primary antibodies were specific for connexin-29 (GJC3, Invitrogen 34-4200, 1:500), TSPAN2 (ProteinTech #20463-1-AP, 1:500), ASPA (ProteinTech #13244-1-AP, 1:500), tetraspanin-28 (CD81, BD Biosciences-US #559517, 1:500), SIRT2 (Abcam #ab67299, 1:500), immunoglobulin superfamily member 8 (IGSF8, Thermo Scientific #PA5-71693, 1:500), CA2 (kind gift from Said Ghandour, 1:1000), MOBP (LS-Bio #LS-C164262/43727, 1:500), MBP (Serotec #PO2687, 1:500), CNP (Sigma #SAB1405637, 1:1000), PLP/DM20, A431 (kind gift from Martin Jung, 1:5000), myelin-associated glycoprotein (MAG, clone 513, Chemicon #MAB1567, 1:500), myelin oligodendrocyte glycoprotein (MOG, clone 8-18C5, kind gift from Christopher Linington 1:500),

claudin-11 (CLDN11, Abcam #ab53041, 1:500), PADI2 (ProteinTech #12110-1-AP, 1:1000), tetraspanin-29 (CD9, Abcam #ab92726, 1:500), alpha-crystallin B chain (CRYAB, ProteinTech #15808-1-AP, 1:500), and PMP2 (ProteinTech #12717-1-AP, 1:500). Appropriate secondary anti-mouse or anti-rabbit antibodies conjugated to HRP were from Dianova (HRP goat-anti-mouse, #115-035-003, 1:5000; HRP goat-anti-rabbit, #111-035-003, 1:5000; HRP goat-anti-rat, #112-035-167, 1:5000). Immunoblots were developed using the Enhanced Chemiluminescence Detection kit (Western Lightning Plus, Perkin Elmer, Waltham, MA) and the Super Signal West Femto Maximum Sensitivity Substrate (Thermo Fisher Scientific, Rockford, IL). Signal was detected using the Intas ChemoCam system (INTAS Science Imaging Instruments GmbH, Göttingen, Germany). Original immunoblots are provided in *Figure 2— source data 1*.

## Immunohistochemistry

Paraffinized human optic nerves were cut into 5 µm sections using the microtome RM2155 (Leica, Wetzlar, Germany) and placed on 1-mm-thick microscope slides (Marienfeld, #1000000, Lauda/ Königshofen, Germany). Immunolabeling of the paraffinized cross sections was performed as follows: sections were incubated for 10 min at 60°C, deparaffinized in a series of incubations in xylol, xylol, xylol/isopropanol (1:1 ratio) for 10 min each, incubated in a series of steps in decreasing ethanol concentration (100, 90, 70, and 50%) for 5 min each, and finally washed in ddH$_2$O for 5 min. Afterward, the sections were incubated for 5 min in 0.01 M citrate buffer (pH 6.0). Then, the sections with citrate buffer were microwaved at 600 W for 10 min. Finally, the slides were left to cool down, rinsed 1 × 5 min with 0.05 M Tris buffer (pH 7.6) containing 2% milk powder, and then blocked with 10% goat serum (Gibco/Thermo Fisher Scientific #16210064, Waltham, MA) diluted 1:4 in PBS (pH 7.4)/1% BSA. Primary antibodies were diluted in PBS/BSA and applied overnight at 4°C. Samples were washed 3 × 5 min in Tris buffer with 2% milk powder (Frema Instant Magermilchpulver, granoVita, Radolfzell, Germany). Secondary antibodies were applied in incubation buffer (1:500 in PBS/BSA) with 4',6-diamidino-2-phenylindole (DAPI, Thermo Fisher Scientific, 1:2000). Slides were then rinsed 1 × 5 min with Tris buffer without milk powder and mounted using Aqua-Poly/Mount (Polysciences, Eppelheim, Germany). Antibodies were specific for PMP2 (ProteinTech #12717-1-AP; 1:200) and human β-Tubulin 3 (TUJ1; BioLegend #MMS-435P; 1:500). Secondary antibodies were donkey anti-mouse Alexa 555 (Invitrogen #A31570, 1:1000) and goat anti-rabbit DyLight 633 (Invitrogen #35562, 1:500). The labeled sections were imaged using the confocal microscope LSM880 (Zeiss, Oberkochen, Germany). The signal was collected with the objective Plan-Apochromat 40×/1.4 Oil DIC M27 using oil (Immersol 518F, Zeiss) and an additional zoom of 1.5. To observe the samples with the light source Colibri (Zeiss), an FS90 filter was used. DAPI was excited at 405, and signal was collected between 431 nm and 495 nm. Alexa 555 was excited with a DPSS 561-10 laser at an excitation of 561 nm, and signal was collected between 571 nm and 615 nm. Then, DyLight 633 was excited with a HeNe633 laser at an excitation of 633 nm and an emission between 647 nm and 687 nm. Finally, the MBS 488/561/633 beam splitter was used to detect Alexa 555 and DyLight 633 and MBS-405 for DAPI, respectively. Images were processed with ImageJ software.

## Retrieval of publicly available scRNA-seq datasets

Eight mouse and six human scRNA-seq datasets published between 2015 and early 2020 were collected for transcriptome analysis. Datasets were selected based on the number of cells designated as oligodendrocytes, and the reported health condition of specimen. Dataset expression matrices and, if available, corresponding metadata were recovered for mouse datasets GSE60361 (*Zeisel et al., 2015*), GSE775330 (*Marques et al., 2016*), GSE113973 (*Falcão et al., 2018*), GSE116470 (*Saunders et al., 2018*), SRP135960 (*Zeisel et al., 2018*), GSE129788 (*Ximerakis et al., 2019*), GSE130119 (*Wheeler et al., 2020*), and GSE140511 (*Zhou et al., 2020*). Human scRNA datasets were retrieved for the Single Cell Portal DroNC-Seq human archived brain (*Habib et al., 2017*), GSE97930 (*Lake et al., 2018*), GSE138852 (*Grubman et al., 2019*), GSE118257 (*Jäkel et al., 2019*), GSE130119 (*Wheeler et al., 2020*), and syn21125841 (*Zhou et al., 2020*). For quality control, each of the retrieved datasets was analyzed using the Seurat R package (version 3.1.4; *Butler et al., 2018*) in an analysis pipeline, including validating sequencing quality, filtering for outlier cells (as specified in *Figure 3—source data 1*), log-normalizing the expression matrix with a scale factor 10,000, high variable gene selection and data scaling, linear dimensionality reduction using principal component analysis (PCA), and

neighboring embedding using UMAP to ensure accurate cell type annotation and detect any potential batch effect. Marker genes used for annotating the oligodendrocyte lineage were *CSPG4*, *PCDH15*, *PDGFRA*, *PTPRZ1*, and *VCAN* for OPCs, *BCAS1*, *ENPP6*, and *GPR17* for NFO, and *CA2*, *CLDN11*, *CNP*, *CMTM5*, *MAG*, *MBP*, *MOBP*, *PLP1*, and *SIRT2* for *MOL*. Specific parameters applied to individual datasets and the number of recovered cells are listed in *Figure 3—source data 1*.

## Merging and integration of scRNA-seq profiles of human and mouse MOL

Cells designated as MOL were subset from each dataset and focused for downstream analysis. Before merging human and mouse datasets, 16255 mouse gene symbols were translated into human gene symbols using a reference gene list from Mouse Genome Informatics (The Jackson Laboratory, version 6.16; retrieved from http://www.informatics.jax.org/downloads/reports/HOM_MouseHumanSequence.rpt on 28 October 2020). 32952 additional mouse gene symbols were translated into human gene symbols by capitalizing the lettering. Gene symbol synchronized human and mouse MOL profiles were first merged and proceeded with the general analysis pipeline for identifying possible batch effects. PCA was performed using the top 2000 most variable genes, and UMAP analysis was performed with the top 20 principal components (PCs); the results implied that the different studies introduced the largest variability for data separation. For integrating all selected human and mouse datasets, the SCTransform (*Hafemeister and Satija, 2019*) pipeline implemented in Seurat was applied. Each dataset underwent SCTransform normalization, and all datasets were integrated using 3000 identified integration features. PCA was conducted downstream and UMAP calculation was performed using the first 20 PCs. Cluster analysis was based on the KNN algorithm calculated with resolution 0.1, and clusters of differentially expressed genes were calculated using the model-based analysis of single-cell transcriptomics (MAST) algorithm (*Figure 4—source data 1*).

## Human and mouse transcriptome correlation analysis

Transcriptome correlation analysis of human and mouse MOL scRNA-seq profiles was performed using the vdW score-transformed average expression of integration features (n = 3000) in each dataset. Subsequently, human and mouse gene average vdW scores were visualized using scatter plot (*Figure 3—figure supplement 2*); Pearson's correlation was calculated for annotated known myelin genes and all genes, respectively.

## GO enrichment analysis

The resulting cluster marker gene lists were input for GO enrichment analysis to detect potential regulated biological processes terms using the gprofiler2 R package (version 0.2.0; *Raudvere et al., 2019*). GO terms with an FDR-corrected p<0.05 were considered as enriched and visualized using EnrichmentMap and AutoAnnotate plugins in Cytoscape (version 3.8.2; *Merico et al., 2010*; *Kucera et al., 2016*; *Shannon et al., 2003*).

## Statistics and reproducibility

Pie charts, heatmaps, and scatter plots were prepared in Microsoft Excel 2016 and GraphPad Prism 9. For the scatter plots, Pearson's correlation and regression line were calculated using GraphPad Prism 9. Relative sample proteomic profile distances were evaluated using Pearson's correlation based on $\log_2$-transformed ppm values. scRNA-seq cluster marker analysis was conducted using MAST algorithm. Data analysis and visualization were performed using GraphPad Prism 9 and R software.

## Acknowledgements

We thank S Ghandour, M Jung, and C Linington for antibodies, A Fahrenholz, D Hesse, R Jung, and B Sadowski for technical help, N Brose for continuous support, and the International Max Planck Research School for Genome Science (IMPRS-GS) for supporting V-IG and SBS. We gratefully acknowledge all contributors of data and data platforms that facilitate access to datasets and promote open-science practices, including AD Knowledge portal, Gene Expression Omnibus (GEO), Mousebrain.org, Single-Cell Portal and the ProteomeXchange Consortium partner repository PRIDE.

# Additional information

## Competing interests

Klaus-Armin Nave: Reviewing editor, *eLife*. The other authors declare that no competing interests exist.

## Funding

| Funder | Grant reference number | Author |
| --- | --- | --- |
| Deutsche Forschungsgemeinschaft | WE 2720/2-2 | Hauke B Werner |
| Deutsche Forschungsgemeinschaft | 2720/4-1 | Hauke B Werner |
| Deutsche Forschungsgemeinschaft | 2720/5-1 | Hauke B Werner |
| European Research Council | MyeliNano | Klaus-Armin Nave |

The funders had no role in study design, data collection and interpretation, or the decision to submit the work for publication.

## Author contributions

Vasiliki-Ilya Gargareta, Josefine Reuschenbach, Sophie B Siems, Lars Piepkorn, Carolina Mangana, Erik Späte, Sandra Goebbels, Wiebke Möbius, Investigation, Writing – review and editing; Ting Sun, Investigation, Supervision, Writing – review and editing; Inge Huitinga, Resources, Writing – review and editing; Klaus-Armin Nave, Writing – review and editing; Olaf Jahn, Conceptualization, Investigation, Writing – review and editing; Hauke B Werner, Conceptualization, Funding acquisition, Supervision, Writing – original draft, Writing – review and editing

## Author ORCIDs

Sophie B Siems http://orcid.org/0000-0002-7760-2507
Ting Sun http://orcid.org/0000-0002-7104-7215
Wiebke Möbius http://orcid.org/0000-0002-2902-7165
Klaus-Armin Nave http://orcid.org/0000-0001-8724-9666
Olaf Jahn http://orcid.org/0000-0002-3397-8924
Hauke B Werner http://orcid.org/0000-0002-7710-5738

## Ethics

Human subjects: Post mortem brain tissue was provided by the Netherlands Brain Bank. Donors gave informed consent to perform autopsy and for the use of clinical and pathological information by researchers, approved by the medical ethics committee of the VU medical center (Amsterdam, The Netherlands) decided by the Tissue Advisory Board with project number 1191. The diagnoses were confirmed by a neuropathologist.

For the procedure of sacrificing vertebrates for preparation of tissue, all regulations given in the German animal welfare law (TierSchG §4) are followed. Since sacrificing of vertebrates is not an experiment on animals according to §7 Abs. 2 Satz 3 TierSchG, no specific ethical review and approval or notification is required for the present work. All procedures were supervised by the animal welfare officer and the animal welfare committee for the Max Planck Institute for Multidisciplinary Sciences, Göttingen, Germany. The animal facility at the Max Planck Institute for Multidisciplinary Sciences is registered according to §11 Abs. 1 TierSchG.

## Decision letter and Author response

Decision letter https://doi.org/10.7554/eLife.77019.sa1
Author response https://doi.org/10.7554/eLife.77019.sa2

# Additional files

## Supplementary files
• Transparent reporting form

## Data availability

The mass spectrometry proteomics data for human myelin are supplied as Figure 1-source data 1 and have been deposited to the ProteomeXchange Consortium via the PRIDE partner repository with dataset identifier PXD029727. Code can be accessed at https://github.com/TSun-tech/Gargareta_etal, (copy archived at swh:1:rev:a8d852183c32a289c5e17905ce2bb29470ffdc2d). Labeled immunoblots are provided in Figure 2-source data 1. Parameters applied for scRNA-seq individual dataset quality control and integrative analysis are provided in Figure 3-source data 1. MAST calculated marker genes from human and mouse integrated MOL subpopulations are provided in Figure 4-source data 1.

The following dataset was generated:

| Author(s) | Year | Dataset title | Dataset URL | Database and Identifier |
|---|---|---|---|---|
| Gargareta VI, Reuschenbach J, Siems SB, Sun T, Piepkorn L, Mangana C, Späte E, Goebbels S, Huitinga I, Möbius W, Nave KA, Jahn O, Werner HB | 2021 | Conservation and divergence of myelin proteome profiles between humans and mice | http://proteomecentral.proteomexchange.org/cgi/GetDataset?ID=PXD029727 | ProteomeXchange, PXD029727 |

The following previously published datasets were used:

| Author(s) | Year | Dataset title | Dataset URL | Database and Identifier |
|---|---|---|---|---|
| Falcão AM, van Bruggen D, Marques S, Meijer M, Jäkel S, ffrench-Constant C, Williams A, Guerreiro-Cacais AO, Castelo-Branco G, Agirre E, Vanichkina D, Floriddia EM | 2018 | Disease-specific oligodendrocyte lineage cells express immunoprotective and adaptive immunity genes in multiple sclerosis | https://www.ncbi.nlm.nih.gov/geo/query/acc.cgi?acc=GSE113973 | NCBI Gene Expression Omnibus, GSE113973 |
| Marques S, Zeisel A, Codeluppi S, van Bruggen D, Falcão AM, Xiao L, Li H, Häring M, Hochgerner H, Romanov RA, Gyllborg G, Muñoz-Manchado AB, La Manno G, Lönnerberg P, Floriddia EM, Rezayee F, Ernfors P, Arenas E, Hjerling-Leffler J, Harkany T, Richardson WD, Linnarsson S, Castelo-Branco G | 2016 | RNA-seq analysis of single cells of the oligodendrocyte lineage from nine distinct regions of the anterior-posterior and dorsal-ventral axis of the mouse juvenile central nervous system | https://www.ncbi.nlm.nih.gov/geo/query/acc.cgi?acc=GSE75330 | NCBI Gene Expression Omnibus, GSE75330 |

*Continued*

| Author(s) | Year | Dataset title | Dataset URL | Database and Identifier |
|---|---|---|---|---|
| Saunders A, Macosko EZ, Wysoker A, Goldman M, Krienen FM, de Rivera H, Bien E, Baum M, Bortolin L, Wang S, Goeva A, Nemesh J, Kamitaki N, Brumbaugh S, Kulp D, McCarroll SA | 2018 | A Single-Cell Atlas of Cell Types, States, and Other Transcriptional Patterns from Nine Regions of the Adult Mouse Brain | https://www.ncbi.nlm.nih.gov/geo/query/acc.cgi?acc=GSE116470 | NCBI Gene Expression Omnibus, GSE116470 |
| Wheeler MA, Clark IC, Tjon EC, Li Z, Zandee SEJ, Couturier CP, Watson BR, Scalisi G, Alkwai S, Rothhammer V, Rotem A, Heyman JA, Thaploo S, Sanmarco LM, Ragoussis J, Weitz DA, Petrecca K, Moffitt JR, Becher B, Antel JP, Prat A, Quintana FJ | 2020 | MAFG-driven astrocytes promote CNS inflammation | https://www.ncbi.nlm.nih.gov/geo/query/acc.cgi?acc=GSE130119 | NCBI Gene Expression Omnibus, GSE130119 |
| Ximerakis M, Lipnick SL, Innes BT, Simmons SK, Adiconis X, Dionne D, Mayweather BA, Nguyen L, Niziolek Z, Ozek C, Butty VL, Isserlin R, Buchanan SM, Levine SS, Regev Aviv, Bader GD, Levin JZ, Rubin LL | 2019 | Single-cell transcriptomic profiling of the aging mouse brain | https://www.ncbi.nlm.nih.gov/geo/query/acc.cgi?acc=GSE129788 | NCBI Gene Expression Omnibus, GSE129788 |
| Zeisel A, Muñoz-Manchado AB, Codeluppi S, Lönnerberg P, La Manno G, Juréus A, Marques S, Munguba H, He L, Betsholtz C, Rolny C, Castelo-Branco G, Hjerling-Leffler J, Linnarsson S | 2015 | Single-cell RNA-seq of mouse cerebral cortex | https://www.ncbi.nlm.nih.gov/geo/query/acc.cgi?acc=GSE60361 | NCBI Gene Expression Omnibus, GSE60361 |
| Zeisel A, Hochgerner H, Lönnerberg P, Johnsson A, Memic F, Zwan JVD, Braun E, Borm LE, Manno GL, Codeluppi S, Furlan A, Lee K, Skene N, Harris KD, Hjerling-Leffler J, Arenas E, Ernfors P, Marklund U, Linnarsson S | 2018 | Molecular Architecture of the Mouse Nervous System | https://www.ncbi.nlm.nih.gov/sra/SRP135960 | NCBI Sequence Read Archive, SRP135960 |

*Continued*

| Author(s) | Year | Dataset title | Dataset URL | Database and Identifier |
|-----------|------|---------------|-------------|-------------------------|
| Zhou Y, Song WM, Andhey PS, Swain A, Levy T, Miller KR, Poliani PL, Cominelli M, Grover S, Gilfillan S, Cella M, Ulland TK, Zaitsev K, Miyashita A, Ikeuchi T, Sainouchi M, Kakita A, Bennett DA, Schneider JA, Nichols MR, Beausoleil SA, Ulrich JD, Holtzman DM, Artyomov MN, Colonna M | 2020 | Human and mouse single-nucleus transcriptomics reveal TREM2-dependent and -independent cellular responses in Alzheimer's disease | https://www.ncbi.nlm.nih.gov/geo/query/acc.cgi?acc=GSE140511 | NCBI Gene Expression Omnibus, GSE140511 |
| Zhou Y, Song WM, Andhey PS, Swain A, Levy T, Miller KR, Poliani PL, Cominelli M, Grover S, Gilfillan S, Cella M, Ulland TK, Zaitsev K, Miyashita A, Ikeuchi T, Sainouchi M, Kakita A, Bennett DA, Schneider JA, Nichols MR, Beausoleil SA, Ulrich JD, Holtzman DM, Artyomov MN, Colonna M | 2020 | Human and mouse single-nucleus transcriptomics reveal TREM2-dependent and -independent cellular responses in Alzheimer's disease | https://doi.org/10.7303/syn21125841 | Synapse, 10.7303/syn21125841 |
| Grubman A, Chew G, Ouyang JF, Sun G, Choo XY, McLean C, Simmons RK, Buckberry S, Vargas-Landin DB, Poppe D, Pflueger J, Lister R, Rackham OJL, Petretto E, Polo JM | 2019 | A single-cell atlas of the human cortex reveals drivers of transcriptional changes in Alzheimer's disease in specific cell subpopulations | https://www.ncbi.nlm.nih.gov/geo/query/acc.cgi?acc=GSE138852 | NCBI Gene Expression Omnibus, GSE138852 |
| Habib N, Avraham-Davidi I, Basu A, Burks T, Shekhar K, Hofree M, Choudhury SR, Aguet F, Gelfand E, Ardlie K, Weitz DA, Rozenblatt-Rosen O, Zhang F, Regev A | 2017 | DroNc-Seq: Single nucleus RNA-seq on human archived brain | https://singlecell.broadinstitute.org/single_cell/study/SCP90/dronc-seq-single-nucleus-rna-seq-on-human-archived-brain | SIngle cell portal; DroNc-Seq: Single nucleus RNA-seq on human archived brain, human-archived-brain |
| Jäkel S, Agirre E, Falcão AM, Bruggen D, Lee KW, Knuesel I, Malhotra D, ffrench-Constant C, Williams A, Castelo-Branco G | 2019 | Altered oligodendrocyte heterogeneity in Multiple sclerosis | https://www.ncbi.nlm.nih.gov/geo/query/acc.cgi?acc=GSE118257 | NCBI Gene Expression Omnibus, GSE118257 |
| Lake BB, Chen S, Sos BC, Fan J, Kaeser GE, Yung YC, Duong TE, Gao D, Chun J, Kharchenko PV, Zhang K | 2017 | Integrative single-cell analysis of transcriptional and epigenetic states in the human adult brain | https://www.ncbi.nlm.nih.gov/geo/query/acc.cgi?acc=GSE97930 | NCBI Gene Expression Omnibus, GSE97930 |
| Jahn O, Siems SB, Kusch K, Hesse D, Jung RB, Liepold T, Uecker M, Sun T, Werner HB | 2020 | The CNS myelin proteome: Deep profile and persistence after post-mortem delay | http://proteomecentral.proteomexchange.org/cgi/GetDataset?ID=PXD020007 | ProteomeXchange, PXD020007 |

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
