## [Editor Report]

In this impressive article, the authors study the similarities and differences between the molecules that comprise the insulation that surrounds human brain nerve fibers (myelin), providing essential insight into how to interpret studies of myelin, from the perspective of different species. In all, this article provides a new resource that will be of interest to the myelin community as well as investigators examining the contributions of oligodendrocytes to human neurodegenerative disease.

---

## [Decision Letter]

**Decision letter after peer review:**

Thank you for submitting your article "Conservation and divergence of myelin proteome and oligodendrocyte transcriptome profiles between humans and mice" for consideration by *eLife*. Your article has been reviewed by 3 peer reviewers, including Kelly Monk as the Reviewing Editor and Reviewer #1, and the evaluation has been overseen by a Reviewing Editor and Gary Westbrook as the Senior Editor. The following individual involved in review of your submission has agreed to reveal their identity: Tara DeSilva (Reviewer #2).

Essential revisions:

1) The biggest suggestion was to clarify how the authors will make this very useful data publicly available. The impact and utility would be greatly increased if a website similar to those associated with, for example, Gerber et al. (*eLife* 2021 Apr 23;10:e58591) was available.

2) Text addition: more details to more clearly explain Figure 3 S2 and its significance in the results.

3) Text addition: clarify the brain regions in the text/methods beyond "white matter".

4) Text addition: the authors note a relative lack of newly-formed oligodendrocytes in humans, and this is stated as a definitive-sounding conclusion, before the study moves on to comparing myelinating oligodendrocytes. Can the authors either speculate more on this apparent difference, if they believe it a true species-relevant difference, or perhaps tone down the conclusion, if it may be experimental context-driven (stages, areas examined etc.)?

5) Text addition: include metabolic support and facilitation of fast impulse propagation as key functions of oligodendrocytes in the introductory section.

(5a) Examine the available data and comment on proteins related to metabolic support.

*Reviewer #2 (Recommendations for the authors):*

This manuscript covers a worthy topic to understand mouse versus the human myelin proteome. These data have far reaching implications for translating studies in preclinical mouse models to human. This study takes a deep dive into understanding differences between mouse and human myelin using both quantitative proteomics and comparing to published scRNA-seq datasets. Important comparisons are presented with relevant proteins known to drive myelin formation. However, an emerging role for oligodendrocytes in providing metabolic support to axons was first presented by both the Nave and Rothstein groups. It would be important to include this in the first sentence of the introduction along with the fact that myelin facilitates salutatory impulse propagation. Since oligodendrocyte metabolic support to axons is now recognized as another essential function, it seems only appropriate to investigate the proteins of interest known to be involved in this pathway in both the proteomics and scRNA-seq data in both the mouse and human. Otherwise, this is a comprehensive study that will likely provide an important resource to the field.

*Reviewer #3 (Recommendations for the authors):*

I do not have any major critiques of this manuscript, and I suspect that the information and comparative analyses presented here will be mined further by the community, investigated in many follow-up studies and that this study will prove to have enormous impact on the field of myelin biology.

With its potential in mind, one aspect that isn't quite clear to me is in what way the data will be made available to the community, as reviewers cannot see the links to the database noted to be present in the letter to the editor.

In any case, it would be spectacular if the authors could create a user-friendly interface that researchers could use to directly search for their proteins of interest and see how they compare across species, etc. etc. A recent manuscript by the Suter group, assessing peripheral nerve molecular states, recently published in *eLife* provides a particularly striking example of a user-friendly interface, which would elevate the impact of this study immeasurably. Apologies if this is already in hand: it wasn't entirely clear from the documents available to review.

In any manuscript like this, one could request validation of yet further observations, or even experimental investigation of interesting differences across species, but I think that the validation presented in the manuscript is already very strong and supports the proteomic methods, which I should note I am not sufficiently expert in to critique in detail. I also think that requesting experimental interrogation of differences would be beyond the scope of this study.

---

## [Author Response]

Essential revisions:1) The biggest suggestion was to clarify how the authors will make this very useful data publicly available. The impact and utility would be greatly increased if a website similar to those associated with, for example, Gerber et al. (eLife 2021 Apr 23;10:e58591) was available.

In response, we have programmed a searchable web interface, which is currently in a test phase and available to the reviewers and editors at https://mpinat-tsun.shinyapps.io/MPINAT_OLG_test/

For two examples (screenshots for two example genes) please see our response to essential revision 5a (below). We are now preparing for long-term hosting of the interface. This will be achieved in the next few weeks, and then the web interface will be accessible via the URL https://www.mpinat.mpg.de/myelin. We have already included this URL in the abstract of our manuscript even though the web interface has not yet been linked to it as of today. Please advise us if this is agreeable. Also kindly let us know if the URL should be placed at another location additionally or instead.

2) Text addition: more details to more clearly explain Figure 3 S2 and its significance in the results.

In response, we have re-written much of the Results section for Figures 3S2-3S3, aiming to resolve the unclarity. However, kindly let us know if further improvement is required.

3) Text addition: clarify the brain regions in the text/methods beyond "white matter".

In response, we now specify the samples as taken from subcortical white matter in the main text and the methods section on “Human samples”. However, we have no further information on the precise location in relation to the specific cortex area (i.e. frontal, parietal, occipital etc).

4) Text addition: the authors note a relative lack of newly-formed oligodendrocytes in humans, and this is stated as a definitive-sounding conclusion, before the study moves on to comparing myelinating oligodendrocytes. Can the authors either speculate more on this apparent difference, if they believe it a true species-relevant difference, or perhaps tone down the conclusion, if it may be experimental context-driven (stages, areas examined etc.)?

In response, we apologize if this sounded like a definite conclusion about the existence of few newly formed oligodendrocytes (NFO) in humans. We have thus modified the text in the Results section, aiming to tone down the relevant sentence and thereby to avoid this impression. In brief, the observation is that the number of cells designated as NFO in the available human datasets (132 cells) is too low for reasonable bioinformatic comparison with those in mice (>10,000 cells). It does not necessarily mean – and we do not wish to imply – that the number of NFO in humans is low in principle. We’d like to suggest two possible main reasons. First, more samples derived from higher ages were analyzed in the human compared to the mouse studies, which may implement a bias against NFO in the human data if a general age-dependent decline of the number of NFO exists in humans. This argument is indirectly supported by the age-dependent decline of the number of cells designated as NFO in mice (see trajectory in Author response image 1). Secondly, the designation of cells as NFO is largely based on the relative abundance of markers originally established in mice. However, these markers may not necessarily be optimal for defining the NFO stage in humans. Yet, upon careful consideration we have decided to keep all designations of cell types and stages as in the original studies from which datasets were utilized. The reason is that we think that a reassessment of the markers defining the NFO stage is beyond the scope of our manuscript as we largely focus on mature oligodendrocytes and myelin. However, we believe that the interesting and relevant issue of the definition and characterization of the NFO stage in the human CNS may be resolved in future work, possibly including the establishment of more suitable marker genes.

**Author response image 1. sa2fig1:** 

5) Text addition: include metabolic support and facilitation of fast impulse propagation as key functions of oligodendrocytes in the introductory section.

We have modified the first sentence of the introduction to include metabolic support

(5a) Examine the available data and comment on proteins related to metabolic support.

This request is not entirely trivial to address considering that hundreds of proteins related to metabolic support are present in the myelin sheath, probably mainly in the non-compact compartments. If agreeable, we would thus like to refer the reviewers and editors to the searchable web interface (see point 1), where all can search for the proteins and transcripts, they are interested in. As two examples, please see Author response image 2 and Author response image 3 for Glutamine synthetase (GS, gene name *Glul*), which was recently shown in experimental mice as an oligodendrocyte-expressed gene relevant for neurotransmission (Xin et al., 2019; PMID: 31116973; DOI: 10.1016/j.celrep.2019.04.094 ), as well as that for sirtuin 2 (SIRT2, gene name *Sirt2*), a deacetylase expressed in oligodendrocytes and myelin (Werner et al., 2007; PMID: 17634366**;** DOI: 10.1523/JNEUROSCI.1254-07.2007 ), which is transferred from oligodendrocytes via extracellular vesicles to myelinated axons (Chamberlain et al., 2021; PMID: 34506725; DOI: 10.1016/j.neuron.2021.08.011 ), in which it is considered to deacetylate mitochondrial proteins, thereby increasing ATP production (Chamberlain et al., 2021; PMID: 34506725; DOI: 10.1016/j.neuron.2021.08.011 ) and preventing neurodegeneration (Fourcade et al., 2017; PMID: 28984064; DOI: 10.1111/acel.12682 ). We feel that it may be beyond the scope of the present manuscript to examine and comment on hundreds of proteins related to metabolic support, in the datasets and hope that checking the searchable web interface is agreeable. Yet, if the reviewers / editors specify particular metabolic pathways or proteins/transcripts, we will examine their expression according to the datasets and comment on them.

**Author response image 3. sa2fig3:** 

Please note that the screenshots were taken from a test version of the searchable web interface. It is quite possible that display options, labeling, links etc may still be optimized before the interface goes online.